# Prevalence and clinical profile of glaucoma patients in rural Nigeria—A hospital based study

Ngozika E. Ezinne[1,2☉], Chukwuebuka S. Ojukwu[2,3☉], Kingsley K. Ekemiri[1‡], Obinna F. Akano[4‡], Edgar Ekure[5‡], Uchechukwu Levi Osuagwu[6,7☉]*

1 Department of Clinical Surgical Sciences, Optometry Unit, University of the West Indies, St Augustine Campus, Trinidad and Tobago, 2 Department of Optometry, Madonna University, Elele campus, Okija, Nigeria, 3 Department of Public Health, Oxford Brookes University, Oxford, United Kingdom, 4 I and Eye Optometry, Bronx, New York, New York, United States of America, 5 Department of Biomedicine, Salus University, Elkins Park, Pennsylvania, United States of America, 6 Translational Health Research Institute (THRI), School of Medicine, Western Sydney University, Campbelltown, NSW, Australia, 7 African Vision Research Institute, Discipline of Optometry, University of KwaZulu-Natal, Durban, South Africa

☉ These authors contributed equally to this work.
‡ These authors also contributed equally to this work
* l.osuagwu@westernsydney.edu.au

**Data Availability Statement:** The data that support the findings of this study are included in the manuscript. In addition, the data required to

## Abstract

### Purpose

To determine the prevalence and clinical presentation of participants with glaucoma attending a public eye care facility in Nigeria.

### Method

Hospital based retrospective study of glaucoma participants aged 50 years and above seen over a 5-year period. Descriptive statistics summarized the demographic, clinical characteristics and treatment of the participants and determined the association of variables with gender and age. Prevalence of the glaucoma by type, and their 95% confidence intervals (CI) were also calculated.

### Result

Of the 5482 case files that were reviewed, 995 (18.15%, 95% CI 17.15–19.19%) had glaucoma particularly primary open angle glaucoma (11.55%, 95%CI 10.73–12.42%) and were mostly females (564, 56.7%) aged 69 ± 12 years (range, 50–103 years). In contrast to other glaucoma types, the prevalence of primary angle closure glaucoma (3.68, 95%CI 3.22–4.22) increased by 15% over 5 years. The mean intraocular pressure ranged from 15–50 mmHg but higher in females than males (27.8 ± 6.1mmHg versus 26.6 ± 6.0 mmHg, $P$ <0.05) who had comparable VA (0.58 ± 0.4 Log MAR) and cup-disc ratios ($P$ >0.05). On presentation, the glaucoma hemi field test (GHFT) was outside the normal limits in 45.5% and 54.5% of males and females, respectively. The type of visual field defect was associated with glaucoma type ($P$ = 0.047). Arcuate scotoma was most common (35.5%) across

replicate all study findings reported in the article are included as supplementary file

**Funding:** The authors received no specific funding for this work.

**Competing interests:** he authors have declared that no competing interests exist.

glaucoma types, paracentral scotoma more common in Secondary glaucoma while Seidel scotoma was highest in NTG (19.3%). Beta-blocker was the mainstay of management (42.2%) but more likely to be prescribed to males while more females received carbonic anhydrase inhibitors.

## Conclusions

The high prevalence of glaucoma in older people remains a public health problem in Nigeria. The fact that about half of the participants presented with visual field defect suggests there is a need for public health messages to emphasize on early glaucoma screening, detection and management.

## Introduction

Glaucoma is a group of disorders characterized by a progressive optic neuropathy resulting in characteristic appearance of the optic disc, and/or irreversible visual field defect that are associated either with elevated intraocular pressure or normal pressure [1]. It is a public health problem accounting for 8% of world blindness and the second leading cause of blindness after cataract [2]. Globally, an estimate of 60.5 million people have glaucoma and about 8.4 million had become blind from the condition [2].

The number of people (aged 40–80 years) with glaucoma has been projected to increase to 111.8 million by 2040 [3,4]. Blindness due to glaucoma can be avoided if the glaucoma is detected early and managed appropriately [5]. The prevalence of glaucoma worldwide is about 1% in older people (aged >50 years) and increases with age [3,6]. A review of relevant population based surveys of glaucoma, visual impairment and blindness in Sub- Saharan Africa indicate that glaucoma affects about 4% adults aged 40 years and above and accounts for 15% of blindness [5]. The prevalence ranges from 0.66% to 1.79% in Eritrea, Liberia, Ghana, South Africa and Malawi [7–9]. Primary open angle glaucoma (POAG) is the most common form of glaucoma among Africans [5] and contributes to 8.4 million cases of bilateral blindness even in developed countries with half of the cases still undiagnosed [10]. In Nigeria, 1,130,000 individuals' ≥40 years are blind and 4.25 million have moderate to severe visual impairment [11].

Various studies [12–14] in different parts of Nigeria have shown that glaucoma is one of the leading causes of blindness in the country and the prevalence is slightly higher in the South-eastern part of the country compared with other regions. In a 1995 population based cross sectional survey conducted in Dambatta local government area (LGA), Kano state, Northwestern Nigeria, the authors reported that 15% of the blindness and 7% of the visual impairment they found, were attributable to glaucoma [15]. Murdoch et al [16] reviewed population based studies published between 1966 to September 2012 on posterior segment eye diseases (PSEDs) in sub-Saharan Africa. They found that in Nigeria, the prevalence of glaucoma was 1.02% in those aged >45 years and noted that African-based studies are needed to help estimate present and future needs and plan services to prevent avoidable blindness.

Nigeria is divided along three main ethnic groups with the Igbos in the Eastern region, Yorubas in the Western region and Hausas in the Northern region. Each ethnic group has its unique culture and the lack of ethnic specific data on sight-threatening diseases such as glaucoma makes it difficult to extrapolate the one group's findings due to differences in cultural and socio-economic activities. There is a need to understand the demographic and clinical presentation of glaucoma in different regions in Nigeria for effective management. Evaluating the

epidemiological and clinical profile of glaucoma patients seen at the Federal Medical Centre Eye clinic Gusau, Zamfara State will shed light on inter-ethnic and regional variations of glaucoma prevalence in Nigeria. It will also provide a useful background information for planning epidemiological surveys on glaucoma in this region as well as other parts of Nigeria with similar socio-demographic and ecological characteristics. Therefore, this study was aimed to assess the epidemiological characteristics and clinical presentations of glaucoma patients' ≥50 years seen at a referral center in Nigeria.

## Materials and methods

### Study setting

This retrospective study of adult participants who attended the glaucoma referral center of the Federal Medical Centre (FMC) Gusau, Zamfara State, Nigeria between, January 2011 and December 2016 (5-year period). The eye clinic is one of the two public/government established eye clinics that serves as a primary health care center for over 3 million residents of Zamfara State and its environs. The region is made up of largely Muslims of Hausa ethnic group many of who (60%) are subsistence farmers that live in rural areas and live in rural areas on less than a dollar per day [17]. There is low literacy level in the region [5,17,18]. Life expectancy in this region is less than 50 years, there is high poverty rate and the region has ill-equipped hospitals and infrastructure in terms of roads, public transport and access to health care services are relatively poor [18].

**Study design and sampling.** This was a hospital-based study of participants diagnosed with glaucoma over 5 years. A non-probability convenience sampling method was utilized because all patients with glaucoma who visited the center during the study period were eligible.

**Inclusion and exclusion criteria.** Data for all participants aged 50 years and over who presented for the first time to this referral center and were diagnosed with glaucoma at the eye clinic during the study period were included. This includes those who had undergone filtration surgery. Participants with ocular hypertension, who did not have changes in optic nerve head or visual function abnormalities were excluded. Also, those with any history of ocular disease that could affect the validity of the ocular fundus examination including macular degeneration, retinitis pigmentosa, hypertensive retinopathy, diabetic retinopathy; those with refractive error of ± 4 Diopter (D) sphere, and/or astigmatism of 3D; participants with significant cataract that affect vision as well as glaucoma participants with incomplete records of C/D ratio, visual field assessment were excluded.

**Techniques for determination of clinical indices of glaucoma.** The hospital does not have electronic records, coding, and database registry. It still operates in hard copy system for storing patients' records' therefore all case files of adult patients diagnosed with glaucoma from 2011 to 2016 were first requested from the hospital administrator and retrieved from the archives with the assistance of the eye clinic department secretary.

Data collection involved the use of a data extraction sheet to extract information on demographics, and clinical profile directly from the patients' files. The data on demographics of patients included gender, age at presentation, ethnic group, religion, and occupation. The clinical profile recorded included presenting visual acuity, intraocular pressure (IOP), vertical cup-to-disc ratios (VCDR), type of glaucoma, glaucoma hemifield test, type of visual field defect and method of management. Visual acuity was measured in Snellen notation and subsequently converted to logMAR notation for the purpose of analysis. Glaucoma hemifield test (GHFT) was performed with automated Humphrey visual field analyzer (Humphrey 740; Carl Zeiss Meditech, Dublin, CA) but global indices including pattern deviation, mean deviation,

pattern standard deviation were not documented in the patients 'files at the time; hence, the global indices were not included in the study. IOP was measured using the Goldmann applanation tonometer mounted on a slit lamp bimicroscope and as a routine practice, were taken between the hours of 8 am to 4 pm when the IOP are most stable [19].

For diagnosis of glaucoma, gonioscopy using a Goldmann 3-mirror and fundus eye exam with the Welch-Allyn (Welch-Allyn Inc., Skaneateles Falls, New York, USA) ophthalmoscope was conducted. The hospital used International Society for Geographical and Epidemiological Ophthalmology (ISGEO) for the diagnosis and classification of glaucoma. Similar diagnosis criteria has been used in other hospital based studies [20–22]. Glaucomatous optic disc atrophy was confirmed by stereoscopic examination of the optic disc with a +90D lens on the slit lamp. A measuring eyepiece graticle (Haag Streit) was used in measuring the vertical optic diameter and cup diameter. Also noted were the presence of notching on the disc rim and any violation of the ISNT rule. The vertical cup-to-disc ratio (VCDR) was used as an index of structural glaucomatous damage. There was no ocular coherence tomography (OCT) in the hospital at the time of data collection, hence retinal nerve fiber layer (RNFL) loss and central corneal thickness (CCT) were not collected.

*Glaucoma diagnosis criteria*. The criteria for the classification of glaucoma at this hospital are described below: Criterion 1 Diagnosis (Structural and Functional Evidence) included eyes with a VCDR of 0.7 or more and less than 0.9 and/or VCDR asymmetry of 0.2 or more or a neuroretinal rim width reduced to less than or equal to 0.1 VCDR (between 11 and 1 o'clock or 5 and 7 o'clock) that also showed a definite visual field defect consistent with glaucoma. Criterion 2 Diagnosis (Advanced Structural Damage With Unproved Field Loss) included participants who could not satisfactorily complete the visual field test but had eyes with VCDR of 0.9 or more and/or VCDR asymmetry of 0.3 or more. Criterion 3 Diagnosis (Optic Disc Not Seen, Field Test Impossible) was given if it was not possible to examine the optic disc, and eyes had visual acuity less than 20/400, presence of relative afferent pupillary defect with IOP of 26 mm Hg or higher, and/or evidence of glaucoma surgery or medical records confirming glaucomatous visual morbidity [23].

**Glaucoma types.** Primary Open Angle Glaucoma (POAG) was defined as open and normal appearing angle with IOP $\geq$21 mmHg associated with either glaucomatous optic disc abnormalities (cupping) or glaucomatous visual field abnormalities or with both. Normal tension glaucoma (NTG) was defined as open and normal appearing angle with IOP $\leq$ 21 mmHg at all times, with glaucomatous optic neuropathy or IOP $\leq$ 21 mmHg at all IOP measurements on record. Primary angle closure glaucoma (PACG) was defined as an eye with an occludable drainage angle and features indicating trabecular obstruction by the peripheral iris, such as peripheral anterior synechiae, irido-corneal contact, elevated intraocular pressure (IOP of 21 mmHg or more), together with evidence of glaucomatous optic nerve damage and visual field (VF) loss. Secondary glaucoma (SG) was defined as raised IOP with glaucomatous optic neuropathy or IOP $\geq$21mmHg associated with positive history and ocular findings of pathologies such as trauma, previous surgery, neovascularization, inflammation, or any other abnormal ocular or systemic findings that could have caused prior or current IOP elevation. In addition, glaucoma, patients with a history of use of topical steroids (6 months), a history of trauma or ocular surgery, chronic uveitis, evidence of pseudo exfoliation or pigment dispersion on slit lamp examination, and those with hyper mature or intumescent cataract were grouped under secondary glaucoma.

**Variables description.** The type of Glaucoma (POAG, NTG, PACG and SG) [24] and the clinical indices of glaucoma were the dependent variables at each time. The changing in frequency of different subtypes of glaucoma was gotten by calculating the total number of people with a particular glaucoma type divided by the total number with glaucoma in that year

multiplied by 100. The independent variables included epidemiological characteristics of age, gender, occupation, ethnic groups and religion and clinical indices including type of VF defect, vertical cup-disc ratio (VCDR), IOP, GHFT, VA in logMAR and treatment of the glaucoma (surgery, medications and combinations). Similar to previous paper [25] and for purposes of analysis, participants with counting finger at 2 feet were considered to have a visual acuity of 2/200 or 20/2000. Those with hand movement at a distance of 2 feet were considered to have an equivalent Snellen acuity of 20/20,000. Also, these were converted to logMAR. Light perception (LP) with or without projection and no light perception (NLP) are not VA measurements but merely the ability to detect a stimulus. Therefore, these factors were excluded from the analysis [26].

**Ethics.**   Approval for this study was obtained from the Institutional Review Board of Madonna University, Nigeria. The study adhered to the tenets of the Declaration of Helsinki and permission to access the patient records was obtained from the management of the Federal Medical Centre (FMC) Gusau, Zamfara State.

**Statistical analysis.**   All data analysis were performed using the IBM SPSS Statistics for Windows, Version 25.0 (IBM Corp., Armonk, NY, USA). Normality distribution of the data was assessed using Kolmogorov–Smirnov test. Data was presented using descriptive statistics using frequencies for categorical variables and mean (±standard deviation, SD; range) for continuous variables. One-way analysis of variance (ANOVA) and chi-square test were performed to assess the differences between groups for the continuous and categorical variables respectively. The differences in the proportion diagnosed with different types of glaucoma by year of diagnosis was also assessed using chi-square test. Univariate analysis was conducted to assess the effects of gender on the clinical indices. The level of statistical significance was set at 5%.

## Results

### Demographic characteristics of the participants with glaucoma

Of the 5482 casefiles of participants aged 50 years and over who attended this hospital over 5 years period, 995 participants were diagnosed with glaucoma. Table 1 presents the characteristics of this study population indicating that nearly all were Muslims, females (56.7%) and of Hausa origin. The mean age of the participants was 69 ± 12 years (mean ±SD), and about 61% were farmers. The clinical indices, glaucoma hemifield test classification, type of visual field defect, glaucoma type and treatment in this study population has been shown in Table 1. The table also shows the mean values for the clinical profiles such as IOP, cup-to-disc ratios, visual acuities and the others.

Of clinical indices, VA was drastically reduced with mean VA of 0.58 ± 0.4 logMAR indicating visual impairment. There were 23 (2.31%) and 6 participants (0.60%) whose VA in either or both eyes respectively was recorded as counting finger (n = 1, 4.3%), hand movement (n = 9, 0.90%), and light perception (15, 1.5%). For 375 participants (37.7%), VA in the better Seeing Eye was worse than 0.5logMAR indicating either low vision (n = 315, 31.6%) or blindness (n = 60, 6.0%) according to the WHO definition for blindness as a best-corrected visual acuity worse than 1.3 logMAR.

The mean IOP in this study group ranged from 15–50 mmHg with an average cup-disc ratio of 0.7. For majority of the participants, beta-blocker was the mainstay of therapy (42.2%) and about 1.8% had glaucoma filtration surgery done. Arcuate and ring scotomas were the predominant visual field defect among the participants consisting of about 58% of the reported visual field defects.

**Hospital prevalence of glaucoma.**   Fig 1 shows the hospital prevalence by glaucoma type over 5 years in this rural referral hospital. Over the five-year study period, 18.15% [95%

**Table 1. Descriptive statistics of measured variables among glaucoma participants.**

| Variables | n (%) |
|---|---|
| **Demography n(%)** | 995/5482 (18.2%) |
| *Age, mean (SD)* | 69.2 (11.8), 50–103 |
| *Gender* | |
| Male | 431 (43.3) |
| Female | 564 (56.7) |
| *Ethnic group* | |
| Fulani | 183 (18.4) |
| Hausa | 631 (63.4) |
| Others | 178 (17.9) |
| *African Traditional* | 9 (1.0) |
| Christian | 84 (8.4) |
| Islam | 901 (90.6) |
| *Occupation* | |
| Employed | 90 (9.0) |
| Farming | 613 (61.6) |
| Retired | 181 (18.2) |
| Self employed | 111 (11.2) |
| **Clinical index**, mean (SD), range | |
| Visual acuity (RE) | 0.58 (0.40), 0–2.8 |
| Visual acuity (LE) | 0.55 (0.38), 0–2.8 |
| Cup-disc ratio (RE) | 0.69 (0.11), 0.30–0.90 |
| Cup-disc ratio (LE) | 0.69 (0.12), 0.3–0.9 |
| Intraocular pressure (RE) | 27 (6), 15–45 |
| Intraocular pressure (LE) | 27 (6), 15–50 |
| **Glaucoma Hemifield Test** | |
| Borderline | 231 (23.2) |
| Outside Normal Limit | 541 (54.4) |
| Reduced Sensitivity | 55 (5.5) |
| Within Normal Limits | 168 (16.9) |
| **Visual field Defects** | |
| Arcuate | 353 (35.5) |
| Paracentral | 52 (5.2) |
| Ring | 224 (22.5) |
| Seidel | 98 (9.8) |
| Tunnel | 268 (26.9) |
| **Glaucoma type** | |
| Normal tension | 57 (5.7) |
| Primary angle closure | 202 (20.3) |
| Primary open angle | 633 (63.6) |
| Secondary | 103 (10.4) |
| **Treatment** | |
| Surgery only | 18 (1.8) |
| Trabeculectomy + Alpha 2 agonist | 49 (4.9) |
| Trabeculectomy + prostaglandin analogues | 10 (1) |
| Trabeculectomy + Beta-blocker | 43 (4.3) |
| Prostaglandin analogue | 112 (11.3) |
| Carbonic anhydrase inhibitor | 78 (7.8) |

(*Continued*)

**Table 1.** (Continued)

| Variables | n (%) |
|---|---|
| Beta blocker | 420 (42.2) |
| Alpha 2 agonist | 265 (26.6) |

VA was recorded in Log MAR = logarithmic minimum angle of resolution; SD = standard deviation; RE = right eye; LE = left eye.

Confidence interval CI 17.15–19.19] had glaucoma in this referral hospital. The highest prevalence was for POAG, which was more than three times higher than that of PACG. The lowest prevalence was for NTG.

**Analysis of glaucoma type.** Chi-square analysis revealed no significant association between the type of glaucoma and the demographic factors of gender ($P = 0.122$), occupation ($P = 0.169$), and ethnic group ($p = 0.408$), but age and year of glaucoma diagnosis were associated with glaucoma type in this study group ($P < 0.0005$, for both). Participants who were diagnosed with NTG were younger ($57 \pm 9$ years) than those in PACG ($71 \pm 11$ years), PAOG ($70 \pm 12$ years), and SG ($69 \pm 12$ years) groups ($P < 0.0005$, for all comparisons).

Fig 2 presents the glaucoma types by year of diagnosis showing that except for PACG, which increased by about 15% over the five-year period, all other glaucoma types showed a decline in the proportion diagnosed over 5 years. Overall, 50% fewer cases were diagnosed with glaucoma in 2016 compared with 2011, in this rural hospital.

The type of visual field defect was also associated with glaucoma type ($P = 0.047$) as shown in Fig 3, with arcuate scotoma (35.5%) being the most predominant visual field defect across all types of glaucoma, followed by tunnel vision. Although fewer people had paracentral scotoma, it was more among those diagnosed with SG and POAG. Seidel scotoma was highest among those diagnosed with NTG (19.3%).

**Analysis of the clinical profiles and treatment types.** The mean values for the clinical profiles by gender is shown in Table 2. The mean IOP ($27 \pm 6$ mmHg) was significantly higher in females than males ($27.8 \pm 6.1$ mmHg versus $26.6 \pm 6.0$ mmHg, $P < 0.05$) who had

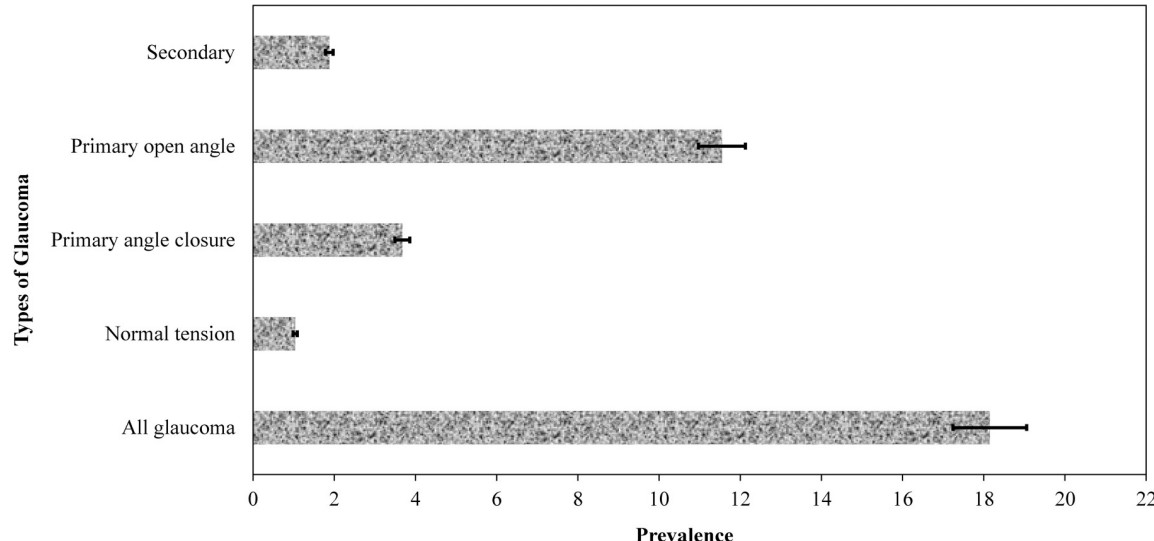

**Fig 1. Prevalence of glaucoma by type over 5 years.** Error bars represent 95% confidence intervals.

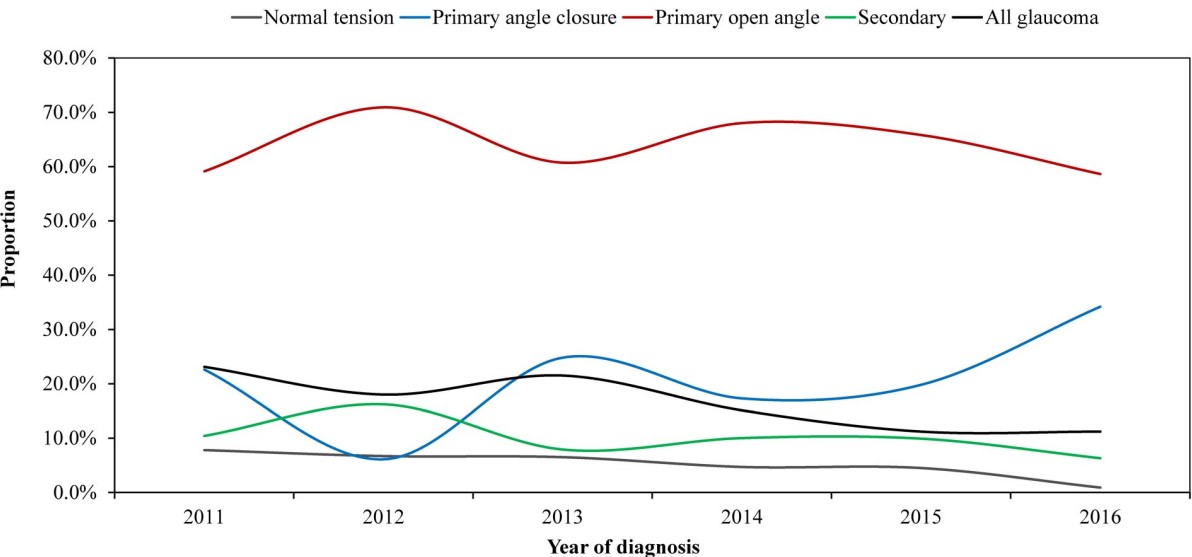

**Fig 2. Percentage distribution of glaucoma type by year of diagnosis.**

comparable VA and cup-disc ratios ($P > 0.05$). For more than half of the participants (n = 541, 54.4%), the glaucoma hemi field test was outside the normal limit and it was within normal limits for 16.9% of the participants (Table 1) and comparable between gender (Table 2, $P = 0.136$).

The treatment type varied significantly between males and females. Males were more likely to be treated with Alpha 2 agonist and beta-blockers, while females were more likely to receive carbonic anhydrase inhibitors (Table 2). About 12.1% of participants had done glaucoma filtration surgery (Trabeculectomy) for control of intraocular pressure and more in males than females (n = 55, 15.1% versus n = 47, 9.8%).

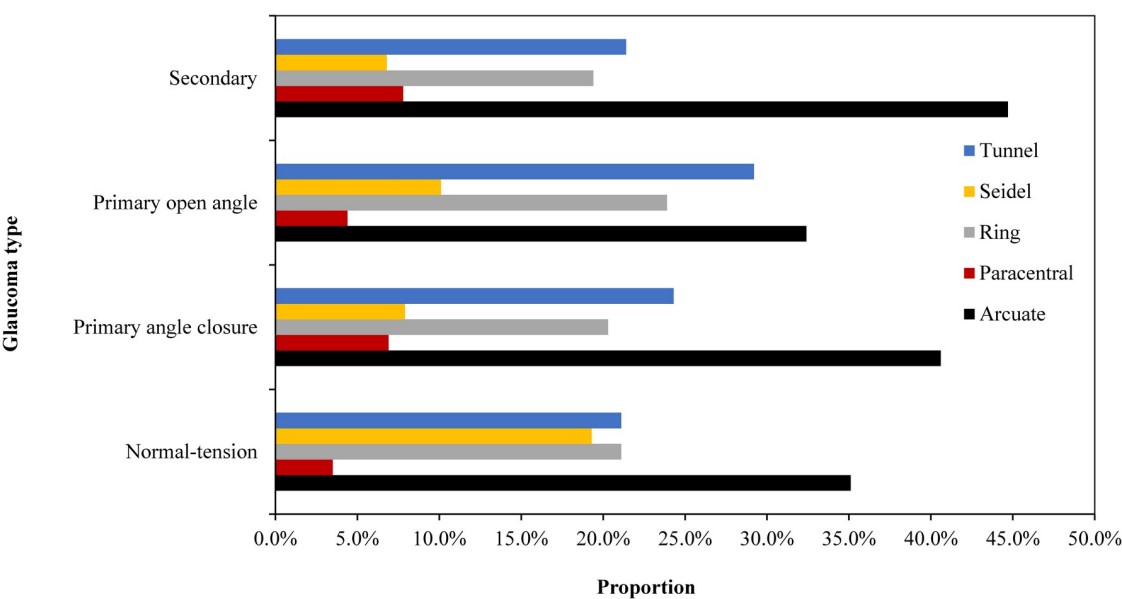

**Fig 3. Percentage distribution of the visual field defect by glaucoma type.**

**Table 2. Clinical indices and treatment of glaucoma participants aged 50 years and over.**

| Variables | Male | Female | *P*-Value |
|---|---|---|---|
| **Clinical index, mean (SD), range** | RE/LE | RE/LE | |
| Visual acuity (RE) | 0.58 (0.42)/0.56 (0.40) | 0.57 (0.39)/0.55 (0.35) | 0.799, 0.661 |
| Cup-disc ratio | 0.68 (0.11)/0.68 (0.11) | 0.69 (0.10)/0.69 (0.12) | 0.268, 0.322 |
| Intraocular pressure (RE) | 26.6 (6.0)/26.3 (5.9) | 27.8 (6.10)/27.4 (5.97) | 0.002, 0.006 |
| **Glaucoma Hemi field Test, n (%)** | | | |
| Borderline | 103 (44.6) | 128 (55.4) | 0.136 |
| Outside Normal Limit | 246 (45.5) | 295 (54.5) | |
| Reduced Sensitivity | 18 (32.7) | 37 (67.3) | |
| Within Normal Limits | 64 (38.1) | 104 (61.9) | |
| **Treatment, n (%)** | | | |
| Trabeculectomy only | 10 (2.3) | 8 (1.4) | 0.021 |
| Trabeculectomy + Alpha 2 agonist | 27 (6.3) | 22 (3.9) | |
| Trabeculectomy + prostaglandin analogues | 4 (0.9) | 6 (1.1) | |
| Trabeculectomy + Beta-blocker | 24 (5.6) | 19 (3.4) | |
| Prostaglandin analogue | 51 (11.8) | 61 (10.8) | |
| Carbonic anhydrase inhibitor | 44 (10.2) | 34 (6.0) | |
| Beta blocker | 169 (39.2) | 251 (44.5) | |
| Alpha 2 agonist | 102 (23.7) | 163 (28.9) | |

VA was recorded in Log MAR = logarithmic minimum angle of resolution; SD = standard deviation; RE = right eye and LE = left eye were for clinical index only.

## Discussion

In the present study, epidemiological and clinical profile of glaucoma patients 50 years and above seen at a health care facility for a period of 5 years were evaluated. There was a high prevalence of glaucoma particularly open angle glaucoma, especially among females, Muslims and farmers. Whereas there was a decline in prevalence for other types of glaucoma, the prevalence of PACG in this underserved community increased by 15% over 5 years. Contrary to a previous report [27], the prevalence of PACG exceeded that of NTG by about 4 folds. The type of visual field defect varied significantly with the glaucoma type but arcuate scotoma was most common in all glaucoma types. Although, beta-blocker was the main drug of choice of glaucoma treatment in this hospital, men were more likely to receive this treatment than women who were more likely to receive carbonic anhydrase inhibitors. At the time of this study, about a quarter of the participants, more men than women (15% versus 10%) already had Trabeculectomy as a surgical procedure for control of their intraocular pressures.

The prevalence of glaucoma reported in this region was considerably higher than previous estimates from survey studies (ranging from 1% to 8.6%) in other parts of the country [14,23,28–30]. Such high prevalence is expected since this region has only two primary health care centers that provide eye care services; therefore high influx of patients will be expected at this center. The fact that our study was in the northern part of Nigeria where majority of the participants were of Hausa ethnic group (less educated) may contribute to the difference in prevalence compared with other studies which included the more educated ethnic groups (Yorubas and Igbos) [23,29]. Also, the lack of awareness and poor utilization of eye care services reported in some parts of Nigeria [31–33] could be the reason for the reduced prevalence recorded. There is a need for more awareness to be created and more eye care outlet established in underserved communities in Nigeria to encourage utilization of eye care services.

Similar to the present report, high prevalence of POAG has been reported in the black race including among African Americans and Afro-Caribbean [5] and in other studies [4,34–39]. It is possible that the prevalence reported in our study may have been underestimated as POAG is usually asymptomatic and people only seek for medical attention when it becomes severe and affect vision. Although the prevalence of POAG observed in this study was higher than previous reports from Nigeria [5,14,18,29,30,40], it was much lower than the 91.2% recorded in another hospital based study from Benin City [41]. Considering the rurality of this community, there is a high possibility that many remain cases of PAOG remain undetected in this population.

The present finding of a significant increase in PACG prevalence during the study period is in agreement with the projected global increase in the prevalence of PACG (from 23–32 million over the next 2 decades [42]. Also, the prevalence of PACG in this Northern hospital exceeds the 1.7% that was reported in Southern hospital studies [14,30]. The study found a marked reduction in the prevalence of all other glaucoma types including POAG, which might not necessarily reflect reduction in glaucoma prevalence but rather a decrease in the utilization of eye care services triggered by insurgency and civil unrest predominant in this region [18]. In addition to these factors, poor awareness of glaucoma and low life expectancy in Nigeria could play a role in the decline in glaucoma prevalence [18,43]. Contrary to our findings, a hospital-based study in Benin City recorded a monthly increase in glaucoma prevalence from 10 to 27% [41] but failed to distinguish between glaucoma types. This increase might be attributed to greater glaucoma awareness, and higher socioeconomic status of the participants since data was from a private owned hospital. However, in another study conducted in a South Korean public hospital, a 54% annual increase in glaucoma prevalence was observed over 5 years. This increase could be attributed to the improvement in glaucoma detection techniques at this hospital, as well as increase in access to eye care services (increased by 9%) and the life span of people in the region (increased by 14.28%) [39].

There are mixed reports on the effect of gender on glaucoma prevalence. The present study found no significant difference in glaucoma prevalence between male and females, which was similar to previous studies from Ghana [44,45]. In contrast, studies from Nigeria [5,40,41], Ghana [46] and South Korea [44] reported a higher prevalence in men than women. Moreover, gender predilection of glaucoma has not been established suggesting the need for more studies to determine the association of glaucoma with gender. Age is a risk factor for glaucoma [47–50] and this was also associated with glaucoma type in this study. Participants with NTG were younger than other glaucoma types even though the overall mean age of participants in this study was similar to previous studies [14,30,41,51–53]. This finding further confirms the importance of visual field and optic nerve assessment as part of the early screening of glaucoma in this population.

The mean VCDR recorded in this study was similar to that of the national eye survey in Nigeria [54], but less than the VCDR recorded among participants in Oyo State Nigeria [23], Tanzania [55] and Netherland [56]. There is a limited information on the distribution of VCDR among Nigerian population; although those from Igbo ethnic group have larger optic disc area and cup than other ethnic groups [5]. The visual field defect, which is one of the hallmark used in the diagnosis of glaucoma, occurs as a result of optic disc cupping. For a good number of the participants in the present study, the glaucoma hemi field test was outside the normal limit field. Uncontrolled IOP due to late presentation could be the reason for the increased visual field loss recorded in this study [20]. Furthermore, the rate of progression of the visual field defect varies in patients, and treatment of the glaucoma may not completely stop the visual field loss as some patients still progress despite treatment. Early screening for glaucoma is highly indicated in this region. Majority of the participants in this study presented

to the clinic at the late stage of glaucoma with many already having significant visual field loss leading to tunnel vision or blindness in at least one eye, which confirms the findings of other studies in Africa [18].

That a good number of participants in this study had severe visual impairment and blindness on first presentation to the clinic was in line with previous reports from Nigeria [5,11,33,40,57] and Saudi Arabia [58]. In North-eastern Nigeria, a study found that about 76% were already blind at hospital presentation. Old age, poor knowledge of glaucoma, rural residence and living far from the hospital were attributed to the late presentation of glaucoma patients in Nigeria [18,57]. In addition, the report of earlier age of onset of glaucoma among Africans or black population may contribute to the high rate of blindness in this population since they would have had the disease for a longer time [40]. Public eye health education and glaucoma screening programs in the rural communities in Nigeria cannot be over emphasized. The Nigerian government should consider ameliorating programs aimed at reducing cost for glaucoma management especially in this region.

The uptake of glaucoma surgery in this region was low and could be attributed to the reported low success rate of Trabeculectomy among black patients [6]. Inadequate access, high cost of surgery, superstition and socio-cultural beliefs may contribute to the preference for medical treatment rather than surgery [59]. In Ethiopia, authors reported a high uptake of glaucoma surgery [60] as ophthalmologists in the country choose surgery over medications due to patients' non-compliance. Similar to a study in Ghana [51], we found that beta-blockers such as timolol were the mainstay of treatment. This could be explained by the fact that it is more affordable and readily available than other classes of drugs including prostaglandin analogues (latanoprost), which are considered the first line of treatment for lowering IOP [61]. In addition, prostaglandin analogues have ocular adverse effects like pruritus, conjunctival hyperemia, ocular irritation, ocular pain, burning, and cilia alteration which may not be pleasant in older people.

## Strengths and limitations

The study has some limitations. First, as a single hospital-based study, the findings are better representatives of the clinical situation compared with population studies but the findings cannot be representative of the general population in Northern Nigeria or the country at large. A population based study is needed with a larger number of patients, to substantiate information obtained from this study. Also, we did not assess associations with other ocular conditions like myopia and comorbid conditions which would require further investigation with additional hospital based data. Retinal nerve fibre layer loss and central corneal thickness were skipped in the diagnosis due to the unavailability of OCT data at the hospital during the period of data collection. Also, global indices were not recorded in the patients' files and this further limits the study. The fact that OCT was not used in the glaucoma diagnosis could have affected the low prevalence of NTG. Normal tension glaucoma (NTG) may be very difficult to detect without OCT and/or pachymetry because it occurs with normal IOP. Despite the limitations, our study is the first to highlight the epidemiology of glaucoma in this region and the key findings were comparable with results from other studies.

## Conclusion

This study found that among people aged 50 years and above in this underserved community, the prevalence of glaucoma was higher than previously reported in other parts of Nigeria. Although primary open angle glaucoma (POAG) showed a decline, it remains a public health problem in Nigeria together with the added burden from the increasing rate of angle closure

glaucoma in this community. The fact that majority of the participants with glaucoma in this region still present late when their ganglion cells and vision have already been severely affected calls for urgent public health measures for glaucoma control in this region. Public health messages emphasizing on early glaucoma screening, detection and management are needed.

## Supporting information

**S1 Data.**
(XLSX)

## Acknowledgments

The authors acknowledge the guidance of Late Prof Alabi, O Oduntan during data collection.

## Author Contributions

**Conceptualization:** Ngozika E. Ezinne, Chukwuebuka S. Ojukwu.

**Data curation:** Ngozika E. Ezinne, Chukwuebuka S. Ojukwu, Kingsley K. Ekemiri, Obinna F. Akano.

**Formal analysis:** Uchechukwu Levi Osuagwu.

**Investigation:** Ngozika E. Ezinne, Kingsley K. Ekemiri.

**Methodology:** Ngozika E. Ezinne, Chukwuebuka S. Ojukwu, Obinna F. Akano, Edgar Ekure, Uchechukwu Levi Osuagwu.

**Software:** Uchechukwu Levi Osuagwu.

**Supervision:** Edgar Ekure.

**Writing – original draft:** Ngozika E. Ezinne, Kingsley K. Ekemiri, Obinna F. Akano, Uchechukwu Levi Osuagwu.

**Writing – review & editing:** Ngozika E. Ezinne, Edgar Ekure, Uchechukwu Levi Osuagwu.

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
