## [Decision Letter · Decision Letter 0]

20 Aug 2021

PONE-D-21-21579

Prevalence and clinical profile of glaucoma patients in rural Nigeria -  A hospital based study

PLOS ONE

Dear Dr. Osuagwu,

Thank you for submitting your manuscript to PLOS ONE. After careful consideration, we feel that it has merit but does not fully meet PLOS ONE’s publication criteria as it currently stands. Therefore, we invite you to submit a revised version of the manuscript that addresses the points raised during the review process.

The manuscript sheds some light and provides useful information about the demographics of glaucoma in Nigeria. However, the manuscript needs meticulous revision regarding the visual field findings to be consistent. In addition, the authors need to address whether he studied population would representative of a single or different ethnic groups.

We look forward to receiving your revised manuscript.

Kind regards,

Ahmed Awadein, MD, Ph.D, FRCS

Academic Editor

PLOS ONE

Journal Requirements:

2. We note that your paper includes detailed descriptions of individual patients/participants. As per the PLOS ONE policy (http://journals.plos.org/plosone/s/submission-guidelines#loc-human-subjects-research) on papers that include identifying, or potentially identifying, information, the individual(s) or parent(s)/guardian(s) must be informed of the terms of the PLOS open-access (CC-BY) license and provide specific permission for publication of these details under the terms of this license. Please download the Consent Form for Publication in a PLOS Journal (http://journals.plos.org/plosone/s/file?id=8ce6/plos-consent-form-english.pdf). The signed consent form should not be submitted with the manuscript, but should be securely filed in the individual's case notes. Please amend the methods section and ethics statement of the manuscript to explicitly state that the patient/participant has provided consent for publication: “The individual in this manuscript has given written informed consent (as outlined in PLOS consent form) to publish these case details.

3. In the ethics statement in the manuscript and in the online submission form, please provide additional information about the patient records used in your retrospective study, including: a) whether all data were fully anonymized before you accessed them; b) the date range (month and year) during which patients' medical records were accessed; c) the date range (month and year) during which patients whose medical records were selected for this study sought treatment. If the ethics committee waived the need for informed consent, or patients provided informed written consent to have data from their medical records used in research, please include this information.

 [Not applicable]. 

Reviewers' comments:

Reviewer's Responses to Questions

**Comments to the Author**

1. Is the manuscript technically sound, and do the data support the conclusions?

Reviewer #1: Partly

Reviewer #2: No

2. Has the statistical analysis been performed appropriately and rigorously? 

Reviewer #1: No

Reviewer #2: Yes

3. Have the authors made all data underlying the findings in their manuscript fully available?

Reviewer #1: Yes

Reviewer #2: No

4. Is the manuscript presented in an intelligible fashion and written in standard English?

Reviewer #1: Yes

Reviewer #2: No

5. Review Comments to the Author

Reviewer #1: This manuscript requires revision in order to make it useful to the reader.

Introduction.

Lines 81-4: An explanation is required as to how this study will shed light on inter-ethnic and regional variations in glaucoma prevalence in Nigeria, as the study was based in a particular hospital (single centre). Was the population multi-ethnic?

Methods. The authors need to state how cases of glaucoma were identified: was there a database, registry, or coding system that allowed such identification, other system employed?

Was VA measured in logMAR directly or in Snellen and subsequently converted? See later as well (lines 76=9)

What type of prevalence was this, and how was it determined (lines 221-4)? As the authors are aware, hospital prevalence is totally different from population prevalence, and have different implications.

How was the changing frequency of different subtypes of glaucoma determined? How do the authors explain the falling frequency of glaucoma over time?

The Discussion could be slightly focused.

Other comments

Line 44, 123.202 and elsewhere: 'hemi-field'

Line 45 and elsewhere: p value with lower case 'p'

Line 576: insert 'and/or' preceding 'irreversible'

Line 111: replace ';' with stop, and commence a new sentence to read 'Further, those with a history....'

Line 120: define 'IOP' at first use in the manuscript (I cant find it)

Line 136: 'VCDR previously defined in line 120; adopt abbreviation subsequently throughout

Line 174: 'logMAR'

Lines 76-9: define these levels of VA in logMAR. Similarly HM, CF and PL have logMAR notations?

Reviewer #2: The authors are not clear with their unit of analysis.they reported that 995 patients with glaucoma were diagnosed over the 5 years of their study and they proceeded to present field defects of 995 patients however they didn't mention which one of the patient was selected for analysis and what criteria was used in the selection of the eye. Furthermore, they stated in line 209 to 211 that 15 patients had visual acuity of light perception in both eyes and that these patients were excluded from analysis (line 177 to 179) then how did they arrive at 995 visual defect results when 15 of the 995 had only light perception bilaterally. Additionally, it is impossible for HVF machine to report GHT results without reporting the global indices.As a glaucoma specialist who has been treating glaucoma patients in Nigeria,I know it is almost impossible to see only 995 glaucoma patients in 5 years.The glaucoma definition the author's quoted in the study is applicable only in prevalent survey not for hospital based study.

6. PLOS authors have the option to publish the peer review history of their article (what does this mean?). If published, this will include your full peer review and any attached files.

Reviewer #1: No

Reviewer #2: No

---

## [Author Response · Author response to Decision Letter 0]

10 Oct 2021

PONE-D-21-21579R1

Prevalence and clinical profile of glaucoma patients in rural Nigeria - A hospital based study Dr Uchechukwu Levi Osuagwu

Dear Dr. Osuagwu,

We've checked your submission and before we can proceed, we need you to address the following issues:

1. Thank you for stating the following financial disclosure: 

 [Not applicable]. 

d) 

e) If you did not receive any funding for this study, please state: “The authors received no specific funding for this work.”

Reply: Done. See cover letter and wording was revised in the manuscript to reflect this. Thanks for changing the online submission on our behalf

2. In the ethics statement in the manuscript and in the online submission form, please provide additional information about the patient records used in your retrospective study, including: 

a) whether all data were fully anonymized before you accessed them; 

Reply: This has been included in the manuscript. 

“…..Only the researcher (CO) has access to identifiable participant data in order to extract the required variables. However, the researcher de-identified the data once the variables were extracted.

 [line 116-118]”

b) the date range (month and year) during which patients' medical records were accessed. 

Reply: This was already in the manuscript lines 100-101 and read: ….January 2011 and December 2016

 If the ethics committee waived the need for informed consent, or patients provided informed written consent to have data from their medical records used in research, please include this information.

Reply: Done. Line 201

3. We note your updated Data Availability Statement: “All relevant data are within the manuscript and additional information can be obtained on reasonable request by the Corresponding author"

Please address the following:

a. We note that you state that all relevant data are within the manuscript. Please confirm at this time whether or not your submission contains all raw data required to replicate the results of your study. Authors must share the “minimal data set” for their submission. PLOS defines the minimal data set to consist of the data required to replicate all study findings reported in the article, as well as related metadata and methods (https://journals.plos.org/plosone/s/data-availability#loc-minimal-data-set-definition).

Reply: The data has been uploaded as supporting information file and revised the availability of data and material section.

“Availability of data and material: The data that support the findings of this study are included in the manuscript. In addition, the de-identified data required to replicate all study findings reported in the article are included as supplementary file”

b. We also note that additional data will be available from the corresponding author. PLOS only allows data to be available upon request if there are ethical, legal, or third-party restrictions on sharing a de-identified data set. Please also note that in the interest of long-term data availability, PLOS data policy states that it is not acceptable for an author to be the sole named individual responsible for ensuring data access. (https://journals.plos.org/plosone/s/data-availability#loc-acceptable-data-access-restrictions)

If there are restrictions, please explain them in detail (e.g., data contain potentially sensitive information, data are owned by a third-party organization, etc.) and who has imposed them (e.g., an ethics committee). Please also provide contact information for a data access committee, ethics committee, or other institutional body to which data requests may be sent. If data are owned by a third party, please indicate how others may request data access.

This information will be helpful in updating your Data Availability Statement. We look forward to hearing from you.

Reply: The statement has been revised and raw data that is required to replicate all results presented have been uploaded as supplementary file. I have also included it in the cover letter.

Kind regards,

Osuagwu Uchechukwu Levi

Corresponding Author

---

## [Editor Report · Decision Letter 1]

26 Oct 2021

PONE-D-21-21579R1Prevalence and clinical profile of glaucoma patients in rural Nigeria -  A hospital based studyPLOS ONE

Dear Dr. Osuagwu,

Thank you for submitting your manuscript to PLOS ONE. After careful consideration, we feel that it has merit but does not fully meet PLOS ONE’s publication criteria as it currently stands. Therefore, we invite you to submit a revised version of the manuscript that addresses the points raised during the review process.

ACADEMIC EDITOR:I cannot find a point-to-point response to reviewers' comments, nor changes in the manuscript in response to these comments. The authors only addressed the changes needed by the journal, but not the reviewers.

We look forward to receiving your revised manuscript.

Kind regards,

Ahmed Awadein, MD, Ph.D, FRCS

Academic Editor

PLOS ONE
---

## [Author Response · Author response to Decision Letter 1]

28 Oct 2021

Rebuttal letter: Response to reviewers’ comments

Dear Editor, thanks for the comments. We had earlier addressed all the reviewers’ comments and those of the editor and we were surprised that you didn’t receive that even though it was in our attached submission. I have again attached the response file for your perusal and highlighted the sections that addressed the reviewers concerns. In addition, we have worked on the recent comment as shown below

New file

PONE-D-21-21579R1

Prevalence and clinical profile of glaucoma patients in rural Nigeria - A hospital based study

PLOS ONE

Dear Dr. Osuagwu,

Thank you for submitting your manuscript to PLOS ONE. After careful consideration, we feel that it has merit but does not fully meet PLOS ONE’s publication criteria as it currently stands. Therefore, we invite you to submit a revised version of the manuscript that addresses the points raised during the review process.

ACADEMIC EDITOR:

I cannot find a point-to-point response to reviewers' comments, nor changes in the manuscript in response to these comments. The authors only addressed the changes needed by the journal, but not the reviewers.

Response: Thanks for the comment. We have addressed all comments in the manuscript and provided a point by point response to all comments in the attached rebuttal letter. It came as a surprise that the reviewer could not find that in the merged document. Below are the responses and all changes in the manuscript have been highlighted in different fonts

Done

Done

Done

Reviewers' comments:

Response: Done

Old file

Response to Reviewer #1 comments: 

This manuscript requires revision in order to make it useful to the reader.

Introduction.

Lines 81-4: An explanation is required as to how this study will shed light on inter-ethnic and regional variations in glaucoma prevalence in Nigeria, as the study was based in a particular hospital (single centre). 

Response 

Various studies [12-14] in different parts of Nigeria have shown that glaucoma is one of the leading causes of blindness in the country and the prevalence is slightly higher in the Southeastern part of the country compared with other regions. In a 1995 population based cross sectional survey conducted in Dambatta local government area (LGA), Kano state, Northwestern Nigeria, the authors reported that 15% of the blindness and 7% of the visual impairment they found, were attributable to glaucoma [15]. Murdoch et al [6] reviewed population based studies published between 1966 to September 2012 on posterior segment eye diseases (PSEDs) in sub-Saharan Africa. They found that in Nigeria, the prevalence of glaucoma was 1.02% in those aged >45 years and noted that African-based studies are needed to help estimate present and future needs and plan services to prevent avoidable blindness. 

Nigeria is divided along three main ethnic groups with the Igbos in the Eastern region, Yorubas in the Western region and Hausas in the Northern region. Each ethnic group has its unique culture and the lack of ethnic specific data on sight-threatening diseases such as glaucoma makes it difficult to extrapolate the one group’s findings due to differences in cultural and socio-economic activities. There is a need to understand the demographic and clinical presentation of glaucoma in different regions in Nigeria for effective management. Evaluating the epidemiological and clinical profile of glaucoma patients seen at the Federal Medical Centre Eye clinic Gusau, Zamfara State will shed light on inter-ethnic and regional variations of glaucoma prevalence in Nigeria. It will also provide a useful background information for planning epidemiological surveys on glaucoma in this region as well as other parts of Nigeria with similar socio-demographic and ecological characteristics. Therefore, this study was aimed to assess the epidemiological characteristics and clinical presentations of glaucoma patients’ ≥50 years seen at a referral center in Nigeria. (Line 73-94).

Reviewer’s comment 

Was the population multi-ethnic?

Response 

No, the population was mostly Hausa because the study was done in the north where majority of the population are Hausas. However, there was still a good proportion of people from other ethnic groups. The relevant information from the study was mainly to highlight the prevalence of glaucoma among Hausa ethnic groups in this region so as to compare with the findings from other regions including studies done in the South East and South West where majority of the population are of Igbo and Yoruba ethnic groups respectively. 

This was also highlighted in the result. The relevant section reads: Table 1 presents the characteristics of this study population indicating that nearly all were Muslims, females (56.7%) and of Hausa origin.

Reviewer’s comment 

Methods. The authors need to state how cases of glaucoma were identified: was there a database, registry, or coding system that allowed such identification, other system employed?

Response 

Done. This has been added to the manuscript for clarification (Line 123-129).

The hospital does not have electronic records, coding, and database registry. The hospital still operates in hard copy system for storing patients’ records’ therefore all case files of adult patients diagnosed with glaucoma from 2011 to 2016 were first requested from the hospital administrator and retrieved from the archives with the assistance of the eye clinic department secretary. Data collection involved the use of a data extraction sheet to extract information on demographics, and clinical profile directly from the patients’ files. 

Reviewer’s comment 

Was VA measured in logMAR directly or in Snellen and subsequently converted? See later as well (lines 76=9).

Response 

All VAs were measured in Snellen and subsequently converted to logMAR. It was previously explained in the old version as stated below: 

for purposes of analysis, participants with counting finger at 2 feet were considered to have a visual acuity of 2/200 or 20/2000. Those with hand movement at a distance of 2 feet were considered to have an equivalent Snellen acuity of 20/20,000. Also, these were converted to logMAR. Light perception (LP) with or without projection and no light perception (NLP) are not VA measurements but merely the ability to detect a stimulus (Line 187-192). 

Reviewers comment 

What type of prevalence was this, and how was it determined (lines 221-4)? As the authors are aware, hospital prevalence is totally different from population prevalence, and have different implications.

Response 

The prevalence of glaucoma reported in this study was a hospital prevalence as only those who attended the hospital and were diagnosed with glaucoma participated in the study. We also made this clear in the previous version of the manuscript (line 113-115) as stated below: 

Data for all participants aged 50 years and over who presented for the first time to this referral center and were diagnosed with glaucoma at the eye clinic during the study period were included. 

We have also made it clearer in the subtitle: Hospital Prevalence of Glaucoma

Reviewer’s comment 

How was the changing frequency of different subtypes of glaucoma determined? 

Response 

The changing in frequency of different subtypes of glaucoma was gotten by calculating the total number of people with a particular glaucoma type divided by the total number with glaucoma in that year multiplied by 100. It has also been added to the manuscript (184-186).

Reviewer’s comment 

How do the authors explain the falling frequency of glaucoma over time?

Response

The study found a marked reduction in the prevalence of all other glaucoma types including POAG, which might not necessarily reflect reduction in glaucoma prevalence but rather a decrease in the utilization of eye care services triggered by insurgency and civil unrest predominant in this region [18]. In addition to these factors, poor awareness of glaucoma and low life expectancy in Nigeria could play a role in the decline in glaucoma prevalence [18, 43]. This was previously explained in the old version of the manuscript (line 316-321). 

Reviewer’s comment 

The Discussion could be slightly focused.

Response 

The discussion has been revised to be more focused. 

Reviewer’s comment 

Other comments

Line 44, 123.202 and elsewhere: 'hemi-field'

Response 

All hemi-field have been corrected to hemifield

Reviewer’s comment

Line 45 and elsewhere: p value with lower case 'p'

Response 

All p value have been changed to P

Reviewer’s comment 

Line 576: insert 'and/or' preceding 'irreversible'

Response 

The revision has been added in the manuscript (Line 57).

Reviewer’s comment 

Line 111: replace ';' with stop, and commence a new sentence to read 'Further, those with a history....'

Response 

The semicolon has been replaced with a full stop (Line 117). 

Reviewer’s comment 

Line 120: define 'IOP' at first use in the manuscript (I cant find it)

Response 

IOP has been defined appropriately at first use as intraocular pressure (IOP) in the manuscript (Line 131).

Reviewer’s comments

Line 136: 'VCDR previously defined in line 120; adopt abbreviation subsequently throughout.

Response 

VCDR abbreviation has been modified throughout.

Reviewer’s comment 

Line 174: 'logMAR'

Response 

LogMar has been changed to logMAR in the manuscript.

Reviewer’s comment

Lines 76-9: define these levels of VA in logMAR. Similarly HM, CF and PL have logMAR notations?

Response 

The section has been revised and the levels of VA has been explained.

Similar to previous paper [25] and for purposes of analysis, participants with counting finger at 2 feet were considered to have a visual acuity of 2/200 or 20/2000. Those with hand movement at a distance of 2 feet were considered to have an equivalent Snellen acuity of 20/20,000. Also, these were converted to logMAR. Light perception (LP) with or without projection and no light perception (NLP) are not VA measurements but merely the ability to detect a stimulus. Therefore, these factors were excluded from the analysis [26].

Reviewer’s comment 

 

Response to Reviewer #2 comments

Reviewer’s comment 

The authors are not clear with their unit of analysis. they reported that 995 patients with glaucoma were diagnosed over the 5 years of their study and they proceeded to present field defects of 995 patients however they didn't mention which one of the patient was selected for analysis and what criteria was used in the selection of the eye.

Response 

All the 995 participants with glaucoma were included in the analysis and no participant was excluded.

Reviewer’s comments 

 Furthermore, they stated in line 209 to 211 that 15 patients had visual acuity of light perception in both eyes and that these patients were excluded from analysis (line 177 to 179) then how did they arrive at 995 visual defect results when 15 of the 995 had only light perception bilaterally. 

Response 

No participant was excluded in the analysis as that was written in error. 

Reviewer’s comment 

Additionally, it is impossible for HVF machine to report GHT results without reporting the global indices. 

Response 

It is true that Humphrey visual field test report GHT and global indices. However, in this hospital it is not common practice for ophthalmologists to record global indices in patients’ files. So we recorded only data found in patients’ files. This limitation has also been included in the manuscript (Line 386-387).

Reviewer’s comment 

As a glaucoma specialist who has been treating glaucoma patients in Nigeria, I know it is almost impossible to see only 995 glaucoma patients in 5 years. 

Response

We have now indicated that this was a hospital prevalence and we agree with the reviewer that it might not be the true reflection of prevalence of glaucoma in Nigeria. However, the prevalence of glaucoma recorded was based on the record given to us by the hospital. 

Reviewer’s comment

The glaucoma definition the author's quoted in the study is applicable only in prevalent survey not for hospital based study.

Response 

The glaucoma definition used was based on the information obtained from the hospital which was confirmed in other studies (line 144-145).

---

## [Editor Report · Decision Letter 2]

22 Nov 2021

Prevalence and clinical profile of glaucoma patients in rural Nigeria -  A hospital based study

PONE-D-21-21579R2

Dear Dr. Osuagwu,

We’re pleased to inform you that your manuscript has been judged scientifically suitable for publication and will be formally accepted for publication once it meets all outstanding technical requirements.

Kind regards,

Ahmed Awadein, MD, Ph.D, FRCS

Academic Editor

PLOS ONE
---

## [Editor Report · Acceptance letter]

24 Nov 2021

PONE-D-21-21579R2 

Prevalence and clinical profile of glaucoma patients in rural Nigeria – A hospital based study 

Dear Dr. Osuagwu:

I'm pleased to inform you that your manuscript has been deemed suitable for publication in PLOS ONE. Congratulations! Your manuscript is now with our production department. 

Kind regards, 

on behalf of

Dr. Ahmed Awadein 

Academic Editor

PLOS ONE